# COLLABORATIVE DATA OPTIMIZATION

## ABSTRACT

Training efficiency plays a pivotal role in deep learning. This paper begins by analyzing current methods for enhancing efficiency, highlighting the necessity of optimizing targets, a process we define as *data optimization*. Subsequently, we reveal that current data optimization methods incur significant additional costs, e.g., human resources or computational overhead, due to their inherently sequential optimization process. To address these issues, we propose COOPT, a highly efficient, parallelized framework designed for collaborative data optimization. COOPT enables participants to independently optimize data subsets, ensuring that the overall performance, once these subsets are collected, remains *comparable* to the sequential optimization of the entire dataset, thus significantly reducing optimization costs for individual participants. Extensive experiments have been conducted on various real-world scenarios to demonstrate the effectiveness and efficiency of COOPT across various datasets and architectures [1].

## 1 INTRODUCTION

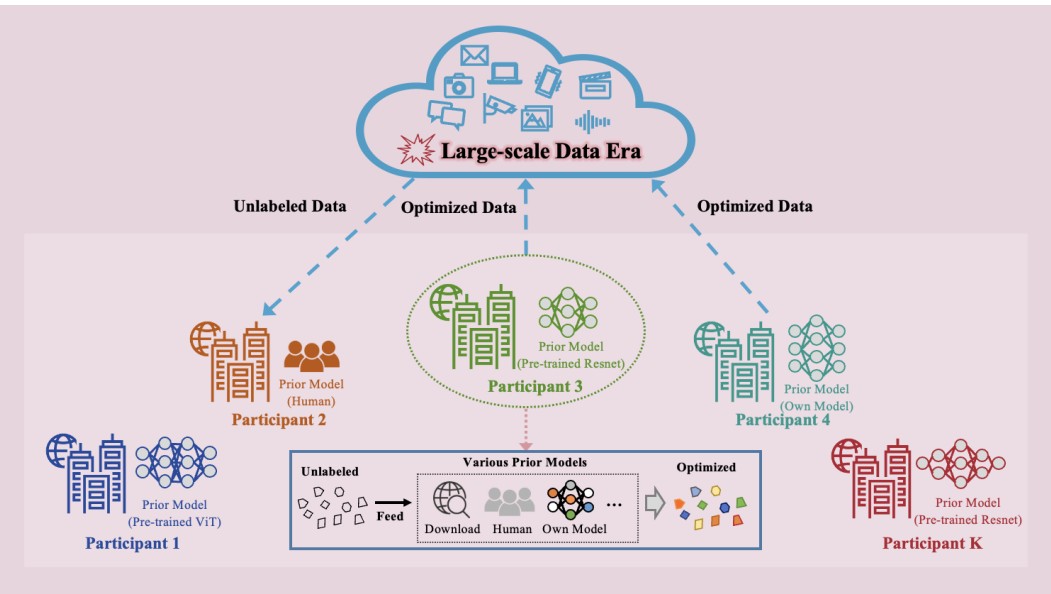

Figure 1: **A Collaborative Data Optimization Framework COOPT.** In practical scenarios involving open-source, large-scale unlabeled datasets, direct utilization via self-supervised learning results in 😣 low training efficiency (Wang et al., 2021). Therefore, we propose COOPT, an 😀 efficient and parallel framework enabling participants to utilize diverse task-agnostic models, such as pre-trained ResNets (He et al., 2016), termed *prior models*, for collaborative data optimization. These prior models can be sourced from internet resources, human expertise, or models trained on the participants' own datasets.

Deep learning has achieved remarkable success across various domains, primarily due to the availability of large-scale, high-quality datasets (Song et al., 2020; Yang et al., 2023). However, despite the abundance of data in the era of big data, a significant portion remains unlabeled (Lei & Tao, 2023).

---

[1]Our code is provided in the Supplementary Materials and will be publicly accessible.

Table 1: **Properties of Various Data Utilization Methods.** 'Optimize $D_X$' and 'Optimize $D_Y$' indicate whether they are optimized. 'SSL' denotes self-supervised learning, 'HA' is human annotation, 'KD' is knowledge distillation, and 'DD' is dataset distillation. '$\mathbf{c}$' denotes the training cost associated with standard supervised learning that employs human-labeled data. '-' is not computable as it is associated with human.

| Method | | Optimize | | Efficiency | | Cost Analysis | Total Efficiency |
| | | $D_X$ | $D_Y$ | Data Optimization | Model Training | | |
|---|---|---|---|---|---|---|---|
| Original | SSL | ✗ | ✗ | 0 | 😠 $(\geq 2\mathbf{c})$ | Extensive Computation | 😠 |
| Data Optimization Methods | HA | ✗ | ✓ | 😠 $(\ -\ )$ | 😐 $(= \mathbf{c})$ | Human Annotation | 😠 |
| | KD | ✗ | ✓ | 😠 $(\geq \mathbf{c})$ | 😊 $(< \mathbf{c})$ | Task-specific Teacher Models | 😠 |
| | DD | ✓ | ✓ | 😠 $(\geq \mathbf{c})$ | 😊 $(< \mathbf{c})$ | Task-specific Pre-trained models | 😠 |
| **CoOpt (Our)** | | ✗ | ✓ | 😊 $(< \mathbf{c})$ | 😊 $(< \mathbf{c})$ | Various Prior Models | 😊 |

Self-supervised learning (Chen et al., 2020b) is proposed to exploit intrinsic relationships within large volumes of unlabeled data to learn meaningful representations. While it reduces dependency on labels, training on such extensive datasets demands considerable computational resources, posing significant computational challenges and diminishing training efficiency (Sun et al., 2024a).

To efficiently leverage unlabeled data, a straightforward but labor-intensive method is through human annotation, thereby transforming them into labeled data. To further enhance training efficiency, methods such as knowledge distillation (Hinton, 2015) and dataset distillation (Wang et al., 2018) have been proposed. Knowledge distillation innovatively leverages soft labels provided by a powerful teacher model to improve the performance of a student model and expedite its training (Dong et al., 2023). Dataset distillation focuses on compressing the original dataset into a smaller subset, enabling a model trained on this distilled data to perform comparably to one trained on the full dataset, thus significantly reducing computational costs.

The data utilization methods discussed above are summarized in Table 1. Essentially, compared to the less efficient self-supervised learning methods, methods achieve higher efficiency by either optimizing targets $D_Y$ through human annotation or pre-trained models (as in knowledge distillation), or by optimizing both input data $D_X$ and targets $D_Y$ (as in dataset distillation). Notably, all these methods necessarily optimize targets $D_Y$, and we term this process as ***data optimization***.

In this paper, we identify the inefficiency of conventional data optimization methods, which suffer from a sequential optimization process with a time complexity ranging from $\mathcal{O}(|D_X|)$ to $\mathcal{O}(|D_X|^2)$, as elaborated in Section 3.2. As a remedy, we propose CoOpt, a highly efficient collaborative framework inspired by crowd-sourcing to achieve parallel data optimization. An overview of CoOpt is depicted in Figure 1, with detailed processes illustrated in Figure 2.

**In summary, our contributions are threefold:**

(a) We propose CoOpt, a highly efficient and parallelized framework for collaborative data optimization. This framework enables participants to independently optimize data subsets, ensuring that when these subsets are collected, the overall performance is *comparable* to sequential optimization of the entire dataset. Therefore, CoOpt significantly reduces optimization costs for each participant.

(b) Within CoOpt, we identify a critical issue: *Target Distribution Inconsistency*, as defined in Section 3.4. This issue arises from the diverse prior models employed by participants, leading to heterogeneity in the target distribution spaces. To address this challenge, we propose an effective target alignment strategy, elaborated in Section 3.5.

(c) Extensive experiments have been conducted across a range of real-world scenarios, involving a variety of prior models trained on *various datasets, architectures, and training paradigms*. Notably, special cases are explored where human or significantly weak models are employed as prior models, verifying the robustness and flexibility of CoOpt. These experiments consistently demonstrate that CoOpt achieves superior effectiveness and efficiency across a range of scenarios.

## 2 RELATED WORK

This section first introduces low-efficiency self-supervised learning (Chen et al., 2020a) on unlabeled data. Subsequently, it reviews existing high-efficiency methods for labeled data, specifically focusing on Knowledge Distillation (Hinton, 2015) and Dataset Distillation (Wang et al., 2018).

## 2.1 Low-Efficiency Self-supervised Learning for Unlabeled Data

To eliminate the need for human annotation, self-supervised learning (Chen et al., 2020b) is proposed to exploit the intrinsic co-occurrence relationships within large volumes of unlabeled data to learn meaningful representations.

Instance-instance contrastive learning has demonstrated effectiveness across various visual classification tasks. For example, InstDisc (Wu et al., 2018) introduces the concept of using instance discrimination as a pretext task. Building on this, CMC (Tian et al., 2020) proposes to use multiple views of an image as positive samples and take another one as the negative. MoCo (He et al., 2020) significantly increases the number of negative samples but utilizes a relatively simplistic strategy for selecting positive samples. Subsequent methods, such as PIRL (Misra & Maaten, 2020), incorporated jigsaw augmentations, and SimCLR (Chen et al., 2020a) highlights the importance of hard positive sample strategies by introducing data augmentation. A notable advancement is BYOL (Grill et al., 2020), which discards negative sampling and surpasses the performance of SimCLR (Chen et al., 2020a). SimSiam (Chen & He, 2021) further investigates the necessity of negative sampling in contrastive representation learning, achieving faster convergence.

**Summary.** *These methods, although not reliant on human annotation, often come with high computational costs due to the need for large batch sizes or memory banks.*

## 2.2 High-Efficiency Data Optimization Methods for Labeled Data

**Knowledge distillation.** Knowledge distillation (Hinton, 2015) innovatively employs soft labels generated by high-capacity teacher models to improve the performance of a student model. Many following works aim to enhance the use of soft labels for more effective knowledge transfer. WSLD (Zhou et al., 2021) analyzes soft labels and distributes different weights for them from a perspective of bias-variance trade-off. DKD (Zhao et al., 2022) decouples the logits and assigns different weights for the target and non-target classes. Moreover, several studies (Yim et al., 2017; Dong et al., 2023) have demonstrated that knowledge distillation can accelerate the optimization process during training.

**Dataset dstillation.** Dataset distillation (Wang et al., 2018) aims to learn a compact distilled dataset that preserves the essential information in the large-scale original dataset, achieving comparable performance to the original dataset with less training time. Current solutions can be categorized based on their optimization mechanisms (Lei & Tao, 2023): meta-learning framework (Wang et al., 2018; Zhou et al., 2022), gradient matching (Zhao et al., 2020; Zhao & Bilen, 2021), distribution matching (Zhao & Bilen, 2023; Yin et al., 2023), trajectory matching (Cazenavette et al., 2022; Guo et al., 2024). Notably, RDED (Sun et al., 2024b) introduces an optimization-free paradigm, which directly crops and selects realistic patches from the original data and then stitches the selected patches into the new images as the distilled dataset.

**Summary.** *Knowledge distillation enhances model training efficiency by optimizing targets $D_Y$, while dataset distillation optimizes both target $D_Y$ and input data $D_X$. Despite their benefits, both approaches are computationally expensive as they require task-specific pre-trained models, which significantly reduces overall efficiency.*

## 3 Collaborative Data Optimization Framework CoOpt

In this section, we begin by formally defining *data optimization* in Section 3.1 . We then analyze and underscore the necessity of our collaborative data optimization framework CoOpt in Section 3.2 . Following this, we provide a comprehensive and detailed description of the proposed CoOpt framework in Section 3.3 . Furthermore, we identify the inherent challenge within this framework in Section 3.4 and present method designed to address the challenge in Section 3.5 .

### 3.1 Data Optimization

As illustrated in Table 1 , we provide a comprehensive comparison of various unlabeled data utilization methods. Self-supervised learning (SSL), which operates without optimizing both targets and input data (indicated by ✗ under 'Optimize $D_X$' and 'Optimize $D_Y$'), is effective but often suffers from low efficiency due to the extensive computational resources required.

In contrast, methods that achieve higher efficiency typically involve optimizing data through different strategies. For example, Human Annotation (HA) provides labeled data (indicated by ✓ under 'Optimize $D_Y$') and achieves high effectiveness and efficiency. However, this approach incurs substantial costs in terms of human resources and time, making it impractical for large-scale datasets or applications requiring rapid deployment. Knowledge distillation (KD) supplies soft labels by teacher models, resulting in high effectiveness and efficiency, yet requires additional computational resources to train the teacher models. Thus, the data optimization process incurs a minimum cost of **c** when only training a teacher model with standard supervised learning, where the cost is **c**.

Dataset Distillation (DD) further extends data optimization by simultaneously optimizing the input samples $D_X$ and the targets $D_Y$. While DD can significantly improve efficiency during the training of new models due to the reduced dataset size, most DD methods rely on complex optimization procedures, such as trajectory matching (Guo et al., 2024), which require models trained on the original datasets to guide the distillation process. This reliance can offset the efficiency gains by introducing additional computational overhead.

**Summary.** *Improving training efficiency necessitates to optimize targets.*

We formally define *data optimization* as the process of optimizing the original dataset $D$ to create an optimal dataset $D'$. The goal is to enable a model $\phi$ trained on $D'$ to achieve comparable performance with *significantly fewer training steps* compared to training on the original dataset $D$.

> **Definition 1 (Data optimization) .** *Data optimization aims to produce optimized data $D'$ s.t.*
>
> $$\mathcal{L}(\phi_{\boldsymbol{\theta}}, D', T') < \mathcal{L}(\phi_{\boldsymbol{\theta}}, D, T) \quad where \quad T' < T, \tag{1}$$
>
> *where $\mathcal{L}$ is the loss function, $T'$ and $T$ denote the training steps required for $D'$ and $D$, respectively, and $\boldsymbol{\theta}$ are the parameters of the neural network $\phi : \mathbb{R}^m \to \mathbb{R}^n$.*

### 3.2 Why Our collaborative data optimization CoOpt is necessary?

> **Proposition 1 (Data optimization with prior model $\psi$) .** *Given samples $D_X = \{\mathbf{x}_i\}_{i=1}^{|D|}$ and an existing prior model $\psi : \mathbb{R}^m \to \mathbb{R}^l$, the objective of data optimization is assigning targets $D_Y = \{\mathbf{y}_i\}_{i=1}^{|D|}$ for the samples to create $D' = \{\mathbf{x}_i, \mathbf{y}_i\}_{i=1}^{|D|}$. We assigns a target $\mathbf{y}_i$ for $\mathbf{x}_i$ as:*
>
> $$D' = \{(\mathbf{x}_i, \mathbf{y}_i) \mid \mathbf{y}_i = \mathbf{W}\psi(\mathbf{x}_i), \forall \mathbf{x}_i \in D_X\}, \tag{2}$$
>
> *where $D_Y$ is the optimized targets, and $\psi(\mathbf{x}_i)$ represents the target of $\mathbf{x}_i$, which means the feature representation. $\mathbf{W} : \mathbb{R}^l \to \mathbb{R}^n$ denotes a random matrix designed to transform the feature vector $\psi(\mathbf{x}_i)$ from dimension $l$ to $n$ without loss of information (Matoušek, 2008). This transformation aligns the output dimension[a] with that required by the model $\phi_{\boldsymbol{\theta}} : \mathbb{R}^m \to \mathbb{R}^n$.*
>
> ___________
>
> [a]Here, $n$ denotes the target dimensionality of $\phi_{\boldsymbol{\theta}}$. In practice, each participant may produce targets of varying dimensions due to the use of different prior models. Therefore, to train the model $\phi_{\boldsymbol{\theta}}$ using the optimized data, we employ the random matrix $\mathbf{W}$ to transform all target vectors to a common dimensionality.

As we discussed above, existing data optimization methods incur substantial costs, as indicated by the 'Extra Cost' in Table 1 . For example, most dataset distillation methods rely on bi-level optimization (Zhao et al., 2020; Kim et al., 2022; Liu et al., 2023), leading to a training cost of $\mathcal{O}(|D|^2)$, where $|D|$ is the number of data samples. To alleviate this computational burden, a promising strategy is to partition the dataset into $K$ splits, thereby reducing the computational cost to $\mathcal{O}(|D|^2/K)^2$. This approach incurs a key question:

> *How can we independently optimize each subset so that, when the subsets are combined, the overall performance is comparable to that achieved by optimizing the entire dataset as a whole?*

Drawing inspiration from Sun et al. (2024a), which demonstrates that employing task-agnostic models for target assignment can accelerate training, we propose to split the data and then independently optimize the targets of each split. Consequently, when the optimized subsets are aggregated, the

___________

[2]This outcome is obtained from $\mathcal{O}(|D|^2/K^2) \times K$, representing $K$ times the processing time for a single partition $\mathcal{O}((|D|/K)^2)$.

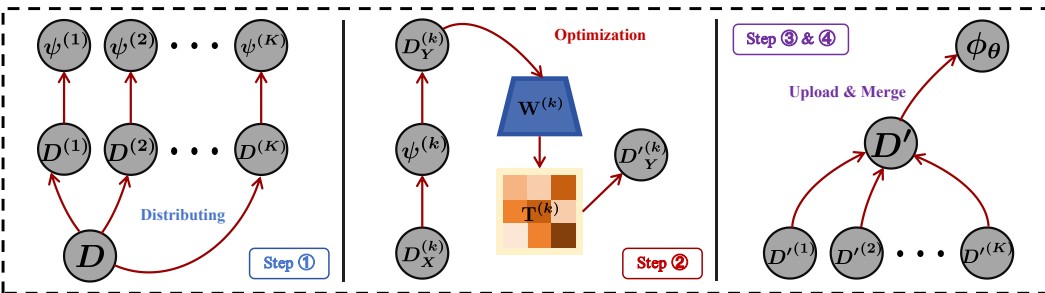

Figure 2: **Lifecycle of the proposed collaborative data optimization framework CoOpt**. The framework encompasses an open data platform and multiple participants, involving four key data operations.

combined targets are the same as those obtained by optimizing the whole dataset, thereby achieving comparable performance. Furthermore, if these $K$ splits are processed in parallel, the computational cost can be further reduced to $\mathcal{O}(N^2/K^2)$. Therefore, **collaboration** among multiple participants becomes essential to distribute the computational burden and enhance computational efficiency. Formally, we define data optimization with a prior model $\psi$ in each participant in Proposition 1.

### 3.3 OVERVIEW OF THE PROPOSED FRAMEWORK CoOpt

CoOpt is a collaborative and parallelized framework that comprises an open data platform and $K$ participants, each equipped with a distinct prior model. Specifically, CoOpt operates through the following four steps:

**Step ①: Data distributing.** The open data platform initiates the process by randomly partitioning the entire set of unlabeled data $D$ into $K$ non-overlapping subsets. Each participant then downloads one of these subsets from the platform, denoted as $D^{(k)}$, where $k$ indicates the $k$-th participant.

**Step ②: Data optimization.** Participants optimize their respective datasets $D^{(k)} = \{\mathbf{x}_i\}_{i=1}^{|D^{(k)}|}$ using their local prior model $\psi^k$. This data optimization process, detailed in Section 3.2, yields optimized targets $D'^{(k)} = \{\mathbf{x}_i, \mathbf{y}_i\}_{i=1}^{|D^{(k)}|}$. However, due to the heterogeneity of prior models among participants, the optimized targets of all participants may exhibit significant variations, leading to divergence in the distribution of the targets. This issue, referred to as *target distribution inconsistency* (defined in Section 3.4), necessitates an alignment strategy to align the target distribution spaces across participants. We propose a solution to this challenge in Section 3.5.

**Step ③: Data uploading.** After optimization, participants upload their optimized datasets $D'^{(k)}$ back to the open data platform.

**Step ④: Data merging.** The platform aggregates all the optimized datasets received from the participants to form a consolidated dataset.

The proposed collaborative parallel process enables participants to independently optimize their subsets while ensuring consistency through the proposed alignment strategies. When combined, the overall results are *comparable* to that achieved by optimizing the entire dataset as a whole. Consequently, this approach markedly reduces individual data optimization costs and enhances data processing efficiency through parallel execution.

### 3.4 AN INHERENT CHALLENGE: TARGET DISTRIBUTION INCONSISTENCY

In our collaborative framework, each participant may employ a distinct prior model, leading to inconsistencies in the target distributions, as illustrated in Figure 4a. For example, participant 1 uses ResNet-18 for optimization, resulting in a target dimension of $512$, while participant 2 utilizes ResNet-50, yielding a target dimension of $2,048$. Such inconsistencies can negatively impact the generalization capabilities of models trained on the optimized data, as they prevent the models from learning representations that are uniformly representative of the overall data distribution.

Considering $N$ participants, participant $k$ uses an prior model $\psi^{(k)}$, resulting in a target distribution space $\mathcal{T}^{(k)}$. Target distribution inconsistency occurs when significant differences exist between these distributions $\mathcal{T}^{(k)}$. The formal definition of target distribution inconsistency is as follows:

**Definition 2 (Target Distribution Inconsistency) .**  *Given a distance metric* $D_{TV}$ *where* $(\mathcal{T}^{(i)}, \mathcal{T}^{(j)}) \in [0, 1], \forall i, j$, *the global inconsistency* $\mathscr{G}$ *among K participants can be quantified as:*

$$\mathscr{G}(\mathcal{T}^{(1)}, \mathcal{T}^{(2)}, ..., \mathcal{T}^{(K)}) = \frac{1}{\binom{N}{2}} \sum_{i=1}^{K-1} \sum_{j=i+1}^{K} D_{TV}(\mathcal{T}^{(i)}, \mathcal{T}^{(j)}), \qquad (3)$$

*where* $D_{TV}$ *is the total variation distance (Verdú, 2014), $i$ and $j$ are participants, and $1/\binom{K}{2}$ is a normalization factor. Target distribution inconsistency exists when $I > \epsilon$, where $\epsilon \in (0, 1)$ is a predefined threshold.*

### 3.5 AN EFFECTIVE STRATEGY: TARGET ALIGNMENT

In the previous section, we identified that the primary challenge arises from the heterogeneity of optimized target distributions across participants. To address this issue, a potential solution is to align the target distributions of all participants' prior models with that of the prior model producing the most optimal target distribution space, referred to as the *best prior model*. Such alignment can be achieved by utilizing an optimizable transformation matrix to map each participant's target distribution to that of the best prior model (Sun et al., 2024a). This alignment strategy ensures consistency across all optimized target distribution spaces.

In summary, *it is crucial to first effectively assess each participant's prior model quality and subsequently train the transformation matrix for alignment.*

**A metric to quantify prior model quality.**   Drawing inspirations from Wang & Isola (2020), which proposes an optimizable metric a.k.a. *uniform value loss* to achieve feature uniformity on the hypersphere during training, we employ this metric to evaluate the quality of prior models. Specifically, each participant downloads a small shared dataset $S_X$ from the platform and computes the uniformity value of their prior model on $S_X$. They then upload this value to the platform, enabling it to determine which participant possesses the best prior model. The uniform value is computed as:

$$\mathcal{V}_{\text{uniform}}(\psi; S) \triangleq \log \mathbb{E}_{\mathbf{x}_i, \mathbf{x}_j \sim S} \left[ e^{\tau \|\psi(\mathbf{x}_i) - \psi(\mathbf{x}_j)\|_2^2} \right], \qquad (4)$$

where $\psi$ is the prior model, $\tau$ is a hyper-parameter set as 2, consistent with Wang & Isola (2020).

A lower uniform value indicates a higher-quality prior model, which optimizes targets of superior quality. Extensive experiments in Figure 3c demonstrate a strong correlation between this metric and the performance of prior, thereby effectively assessing the quality of targets.

**Alignment.**   Upon identifying the best prior model, all participants, excluding the best prior model itself, proceed to train an optimizable transformation matrix. Specifically, the participant owning the best prior model disseminates its optimized targets, denoted as $S_Y{}^\star$, computed over the shared dataset $S_X$ using prior model $\psi^\star$, to facilitate the alignment of target distribution spaces for other participants. Subsequently, each participant $k$ optimizes a lightweight transformation matrix, denoted as $\mathbf{T}^{(k)}$, on the shared dataset $S_X$. The optimization problem is defined as follows:

$$\mathbf{T}^{(k)} = \arg\min_{\mathbf{T} \in \mathbb{R}^{n \times n}} \{ \|\mathbf{T} \cdot \psi^{(k)}(\mathbf{S}_X) - \mathbf{S}_Y{}^\star\|_2^2 \}, \qquad (5)$$

where $\mathbf{S}_X$ represents the matrix form of $S_X$, suitable for input into the network $\psi^{(k)}$, and $\mathbf{S}_Y{}^\star$ also represents the matrix form of $S_Y{}^\star$. After obtaining the transformation matrix $\mathbf{T}^{(k)}$, the participant can convert the optimized targets for its own data using this matrix: $D_Y{}^{(k)} = \mathbf{T}^{(k)} \cdot \psi^{(k)}(\mathbf{D}_X{}^{(k)})$, where $\mathbf{D}_X{}^{(k)}$ denotes the participant's subset, and $D_Y{}^{(k)}$ are the adjusted targets aligned with the best prior model's target distribution space. As illustrated in Figure 4b, the proposed alignment strategy effectively mitigates target distribution space inconsistency.

## 4 EXPERIMENTS

In this section, we outline the experimental setup and assess the performance of our proposed collaborative data optimization framework, COOPT, across various real-world scenarios, employing different datasets and architectures, as discussed in Section 4.2 . Subsequently, we explore a continuous data optimization framework that allows the prior models of participants to evolve, thereby further optimizing the targets, as detailed in Section 4.3 . Finally, we validate the effectiveness of the introduced uniform value metric and the target alignment strategy in Section 4.4 .

## 4.1 Experimental Setup

**Datasets and Networks:** We conduct experiments on both large-scale and small-scale datasets, including Tiny-ImageNet ($64 \times 64$) (Le & Yang, 2015), CIFAR-100 (Krizhevsky et al., 2009a) and CIFAR-10 ($32 \times 32$) (Krizhevsky et al., 2009b). Following previous self-supervised studies (He et al., 2020; Chen et al., 2020a; Grill et al., 2020; Chen & He, 2021; Assran et al., 2023; Zhang et al., 2024), we employ a range of model capacities backbone architectures to evaluate the generalizability of our method, including ResNet-18, 50, 101 (He et al., 2016), ViT (Dosovitskiy et al., 2020).

**Baselines:** For the unlabeled data utilization, referring to a prior widely-used benchmark (Da Costa et al., 2022), we consider several state-of-the-art self-supervised methods as baselines for a broader practical impact, including: SimCLR (Chen et al., 2020a), BYOL (Grill et al., 2020), DINO (Caron et al., 2021), MoCo (He et al., 2020), SimSiam (Chen & He, 2021), and SwAV (Caron et al., 2020).

**Evaluation and Metrics:** Following previous benchmarks and research (He et al., 2020; Chen et al., 2020a; Grill et al., 2020; Chen & He, 2021), we evaluate the test accuracy (%) of all the trained models using offline linear probing strategy to reflect the representation ability of the trained models, and ensure a fair and comprehensive comparison with baseline approaches. Additionally, we measure the computational efficiency by evaluating the time cost (s).

**Implementation details:** In this study, we introduce a collaborative data optimization framework, COOPT, which involves an open data platform and multiple participants. In practical applications, *each participant can use publicly pre-trained models or their own models directly as the prior model.* To simulate the diversity of prior models in practical applications, we train multiple prior models for participants across three key dimensions:

- *Training Paradigm:* Models are trained using various paradigms, such as supervised learning and self-supervised learning. Specifically, for supervised learning, we employ cross-entropy loss, while for self-supervised learning, we primarily utilize the BYOL framework (Grill et al., 2020).
- *Prior Dataset:* These prior models of participants are trained on extensive and public datasets, including CIFAR-10/100, Tiny-ImageNet, and ImageNet-1k.
- *Architecture:* Popular architectures such as ResNet and Vision Transformer (ViT) are employed.

We train the 16 different models based on a widely recognized supervised learning and self-supervised learning open-source benchmark (Da Costa et al., 2022). For the model trained on optimized data, we use the AdamW optimizer, the same as baselines. The size of mini-batch is set as 128. For all experiments, we utilize three random seeds and report both the mean and variance of the results.

## 4.2 Comparison with the state-of-the-art methods

We evaluate our framework, COOPT, across various scenarios: (1) participants use a diverse range of prior models trained on different datasets, architectures, and training paradigms; (2) we specifically evaluate the robustness of our method when employing different prior datasets used for training prior models, as well as (3) varying scalar networks; (4) finally, we explore special cases involving human or weak models, particularly where there are resource-rich or resource-poor participants.

**A Diverse Range of Prior Models.** Given the diversity among participant models, we employ a comprehensive set of 16 models, as detailed in Section 4.1 . These models are trained using *various training paradigms, datasets, and architectures.* Details of the training processes for these models are provided above. As presented in Table 2 , it is evident that *our method* COOPT *demonstrates superior performance and training efficiency compared to existing self-supervised learning methods.*

(a) In terms of performance, the proposed COOPT achieves results comparable to or exceeding state-of-the-art self-supervised learning techniques. For instance, it achieves an improvement of 3.3% over the leading self-supervised approach BYOL on Tiny-ImageNet. Notably, COOPT demonstrates more significant improvements on larger-scale datasets, such as Tiny-ImageNet, which is particularly advantageous given the current emphasis on large-scale data era.

(b) Regarding training efficiency, COOPT demonstrates a substantial reduction in training costs across various datasets. In particular, on the large-scale Tiny-ImageNet dataset, COOPT achieves a training speed that surpasses BYOL and SwAV by a factor of approximately $\times 2.48$ and $\times 1.94$.

**Diverse Prior Datasets.** In the last experiment, we employ prior models that have been pre-trained on a variety of datasets, some of which are congruent with the unlabeled training data, while others

Table 2: **Comparison of CoOpt with Various Self-Supervised Learning Methods Accuracy (%) and Training Time (s).** Evaluations are conducted on four datasets: CIFAR-10 (CF-10), CIFAR-100 (CF-100), and Tiny-ImageNet (T-IN). The best results are highlighted in **bold**. ↑ indicates the *performance* improvement over the second-best results. × denotes the factor of *training speed* compared to the second-best results.

| Dataset | Metric | BYOL | DINO | MoCo | SimCLR | SimSiam | SwAV | CoOpt |
|---------|--------|------|------|------|--------|---------|------|-------|
| CF-10 | Acc. (%) | $82.8 \pm 0.1$ | $82.6 \pm 0.0$ | $82.9 \pm 0.1$ | $83.1 \pm 0.0$ | $79.0 \pm 0.0$ | $82.9 \pm 0.1$ | **83.5 ± 0.1** (↑ **0.4**) |
| | Time (s) | 1376.56 | 1457.22 | 1349.56 | 1114.81 | 1090.79 | 1012.74 | **540.43** (× **1.87**) |
| CF-100 | Acc. (%) | $51.7 \pm 0.1$ | $51.0 \pm 0.0$ | $57.8 \pm 0.1$ | $55.4 \pm 0.0$ | $44.6 \pm 0.1$ | $53.2 \pm 0.1$ | **59.4 ± 0.0** (↑ **1.6**) |
| | Time (s) | 1406.17 | 1419.69 | 1425.80 | 1103.45 | 1139.14 | 1072.44 | **548.11** (× **1.95**) |
| T-IN | Acc. (%) | $43.9 \pm 0.2$ | $36.1 \pm 0.0$ | $42.4 \pm 0.2$ | $41.5 \pm 0.1$ | $40.8 \pm 0.0$ | $39.9 \pm 0.1$ | **47.2 ± 0.1** (↑ **3.3**) |
| | Time (s) | 7086.62 | 7030.90 | 7133.98 | 5621.33 | 5531.92 | 5540.96 | **2852.6790** (× **1.94**) |

Table 3: **Comparison of CoOpt with BYOL Across Diverse Prior Datasets.** For instance, "CF-10 (P)" indicates participants' prior models are trained on CIFAR-10. **Bold** means the best results. Underline indicates the results when the prior dataset is identical to the training data. All models are based on ResNet-18 architectures.

| Dataset | BYOL (Baseline) | Our CoOpt (Diverse Prior Datasets) | | | |
|---------|-----------------|-------------|--------------|------------|------------|
| | | CF-10 (P) | CF-100 (P) | T-IN (P) | IN-1K (P) |
| CF-10 | $82.8 \pm 0.1$ | $86.6 \pm 0.0$ (↑ 3.8) | $80.9 \pm 0.0$ (↓ 1.9) | $81.6 \pm 0.1$ (↓ 1.2) | **88.1 ± 0.0** (↑ **5.3**) |
| CF-100 | $51.7 \pm 0.3$ | $54.9 \pm 0.1$ (↑ 3.2) | $60.0 \pm 0.1$ (↑ 8.3) | $56.8 \pm 0.0$ (↑ 5.1) | **63.7 ± 0.0** (↑ **12.0**) |
| T-IN | $43.9 \pm 0.2$ | $38.3 \pm 0.0$ (↓ 5.6) | $40.2 \pm 0.1$ (↓ 3.7) | $49.0 \pm 0.0$ (↑ 5.1) | **55.8 ± 0.1** (↑ **11.9**) |

are incongruent. We refer to these datasets as the *prior datasets*. To rigorously evaluate the influence of these prior datasets, we perform an analysis across scenarios where the prior datasets used for prior models either align with or differ from the unlabeled training dataset. For instance, in the aligned scenario, the training dataset is CIFAR-10 (CF-10), and the prior models are also trained on CIFAR-10 (CF-10 (P)). Conversely, in the divergent scenario, the training dataset remains CIFAR-10, while the prior models are trained on CIFAR-100 (CF-100 (P)). We conduct experiments on four public datasets, with results detailed in Table 3.

(a) Prior models trained on the same dataset as the training data can yield significant improvements.
(b) For complex training datasets, using simpler prior datasets may degrade performance compared to BYOL, as they provide less informative guidance.
(c) However, for all unlabeled training datasets, employing prior models trained on ImageNet-1K can result in substantial improvements, owing to their robust generalization capabilities. This is especially pertinent in practical applications, given that the majority of pre-trained models accessible in internet resources are derived from ImageNet-1K.

**Diverse Architectures of Prior Models.** To further verify the robustness of CoOpt across various prior model architectures, we perform experiments on various datasets using a diverse range of networks to train prior models. This includes large-scale networks such as ResNet-101 (He et al., 2016) and Swin-V2-Tiny (Liu et al., 2021), as well as smaller-scale networks like ResNet-18 (He et al., 2016), EfficientNet-B0 (Tan & Le, 2019) and MobileNet-V2 (Sandler et al., 2018). The results, presented in Table 4, demonstrate that CoOpt *consistently achieves superior performance across various architectures.*

Table 4: **Comparison of CoOpt with BYOL Across Diverse Architectures of Prior Models.** We use both large-scale and small-scale networks for prior models. **Bold** means the best results.

| Dataset | BYOL | CoOpt |
|---------|------|-------|
| CF-10 | $82.8 \pm 0.1$ | **87.5 ± 0.2** |
| CF-100 | $57.4 \pm 0.1$ | **63.8 ± 0.1** |
| T-IN | $43.9 \pm 0.2$ | **55.7 ± 0.1** |

**Extreme Cases of Prior Models: Human or Weak.** In real-world applications, extreme cases arise due to the varying capabilities of participants. For example, some participants have extensive resources and employ human annotators for labeling, while others may have limited resources and rely on weak models with inferior performance and generalization abilities. In our experiments, we define weak models as those trained during intermediate stages that are even far from convergence.

Table 5: **Comparison of CoOpt with BYOL in Presence of Human or Weak Prior Models.**

| | Prior Models | | Dataset | |
|--------|-------|------|---------|---------|
| Method | Human | Weak | CF-10 | CF-100 |
| BYOL | – | – | $82.8 \pm 0.1$ | $57.4 \pm 0.1$ |
| CoOpt | ✗ | ✗ | $83.5 \pm 0.1$ | $59.4 \pm 0.0$ |
| | ✗ | ✓ | $83.3 \pm 0.1$ | $58.7 \pm 0.1$ |
| | ✓ | ✗ | **86.7 ± 0.0** | **61.0 ± 0.0** |

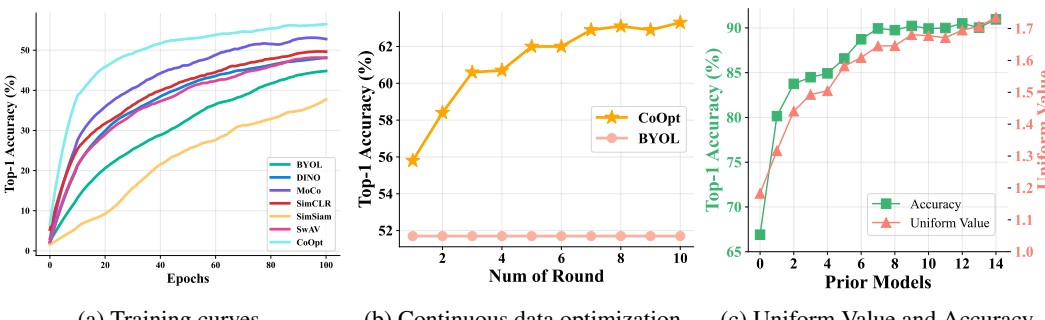

|(a) Training curves. | (b) Continuous data optimization. | (c) Uniform Value and Accuracy.|

Figure 3: **(a) Training curves of various self-supervised learning methods, including our proposed COOPT.** It is evident that our COOPT demonstrates superior performance compared to the other methods. **(b) Practical Scenario:** The prior models of each participant experience temporal evolution, leading to improved target quality across multiple rounds. **(c) Correlation Verification:** Examine the relationship between the uniform value and performance, which demonstrates a strong negative correlation, quantified by $\rho = -0.9714$.

To simulate the conditions, in addition to the 16 diverse prior models detailed in Section 4.1, we incorporate 10 prior models, either human or weak models. The results, presented in Table 5, clearly indicate that *our method,* COOPT*, maintains robustness despite the incorporation of weak models.* Furthermore, the integration of high-capacity human models leads to significant improvements.

### 4.3 CONTINUOUS DATA OPTIMIZATION

We explore another practical scenario where the prior models of each participant experience temporal evolution. For instance, an initially prior model of a participant, such as ResNet-18, may be evolved to a higher-capacity model like ResNet-50 as the participant's resources improve. Consequently, within COOPT, data optimization can be a continuous process.

We simulate this scenario by conducting experiments where, in each interaction round between the platform and participants, 20% of the participants randomly update their model architectures to reflect an increase in model capacity. The training curves across 10 rounds on CIFAR-100 are depicted in Figure 3b. The results have demonstrated that *in* COOPT*, as the prior models evolve, the quality of the targets improves, thereby facilitating continuous optimization.* In particular, compared to the baseline BYOL, these improvements can achieve an enhancement of 11.6%.

### 4.4 ABLATION STUDY

**Effectiveness of Uniform Value.** To evaluate the effectiveness of uniform value in estimating the quality of prior models, we employ a diverse set of prior models, calculating both their uniform value and test accuracy. We employ the Spearman rank correlation coefficient[3] $\rho$ (Zar, 2014) to quantify the association between Uniform Value and accuracy. As illustrated in Figure 3c, the Spearman rank correlation coefficient is $\rho = -0.9714$, *indicating a strong correlation between uniform value and model performance, thereby effectively assessing the quality of prior models.*

**Target Distribution Alignment.** In real-world applications, participants often employ diverse prior models, which results in target distribution space inconsistencies, as demonstrated in Figure 4a. We conduct an ablation study w/o alignment and w/ alignment to verify the importance of target distribution space alignment, and the training curves presented in Figure 4d.

(a) Compared to without alignment (green line), our proposed approach (blue line) yields a performance improvement of 16.9%, which highlights the critical importance of aligning the target distribution space. Notably, the training curve without alignment (green line) initially ascends before declining, suggesting that during the early phase of model training, optimizing the target leads to performance enhancement. However, as training continues, the severely inconsistent targets significantly degrade model performance.

---

[3]The formula is: $\rho = 1 - \frac{6\sum d_i^2}{n(n^2-1)}$, where $d_i$ represents the difference between the ranks of each pair of observations and $n$ denotes the number of observations.

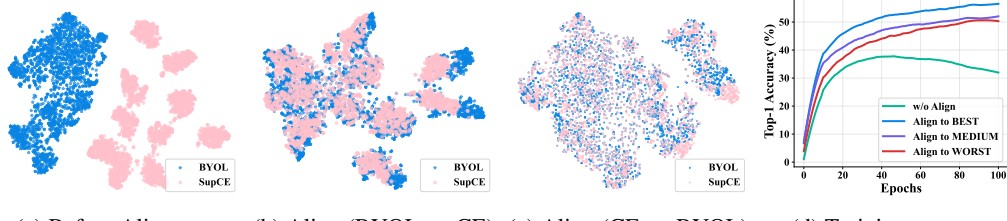

(a) Before Alignment  (b) Align (BYOL → CE)  (c) Align (CE → BYOL)  (d) Training curves.

Figure 4: **(a), (b), (c): Visualization of t-SNE for optimized targets generated by two distinct models (BYOL (acc. = 82%) and SupCE (acc. = 90%).).** Aligning to the worse model (c) results in diminished target quality. (d) **Training curves with and without alignment**, demonstrating the importance of alignment.

(b) Furthermore, we compare our alignment strategy, which aligns to the best prior model, with two other straightforward strategies: aligning to a medium prior model (purple line) and aligning to the worse prior model (red line). It is evident that all three alignment strategies outperform the scenario without alignment, and *aligning to the best prior model provides the most significant performance gains.*

(c) To further analyze the underlying reasons, we employ t-SNE visualization. Notably, a comparison between alignment with the better participant ( Figure 4b ) and the worse participant ( Figure 4c ) reveals that alignment with a worse-quality model diminishes the representative capability of the targets. This, in turn, results in less effective guidance for model training.

**Influence of Shared Unlabeled Data Size.** The shared unlabeled data $S$ is employed to estimate the uniform value and compute the transformation matrix necessary for target alignment, as detailed in Section 3.5 . To explore the influence of the size of shared unlabeled data, we conduct experiments on both CIFAR-10 and CIFAR-100 datasets, varying the size of the shared data in proportions of $\{0.01, 0.05, 0.1, 0.2, 0.4, 0.6, 0.8\}$ relative to the original dataset. The results are presented in Figure 5 . It is observed that *as the size of the shared data increases, performance gains become marginal.* Specifically, increasing the proportion of data from $0.01\%$ to $0.05\%$ results in improvements of $2.4\%$ and $2.5\%$ for CIFAR-10 and CIFAR-100, respectively. However, increasing from $0.05\%$ to $0.8\%$ This suggests that a small-sized shared unlabeled dataset is adequate for accurate uniform value estimation and the computation of the transformation matrix.

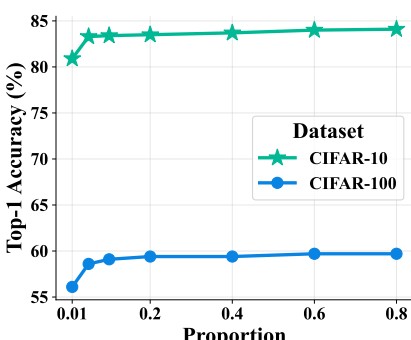

Figure 5: **Influence of shared data size.** As the size of the shared data increases, the performance gains diminish A small dataset size is sufficient to estimate uniform value and compute the transformation matrix.

## 5 CONCLUSION AND FUTURE WORK

**Conclusion.** In this paper, we introduce COOPT, an efficient and highly parallelized framework for collaborative data optimization. This framework enables participants to independently optimize data subsets such that, when aggregated, the overall performance is comparable to that achieved by sequentially optimizing the entire dataset. Consequently, COOPT significantly reduces individual data optimization costs. Within COOPT, we identify a critical issue: Target Distribution Inconsistency, which arises from the diversity of prior models used in data optimization. To mitigate this, we propose an effective target alignment strategy. Extensive experiments conducted across various real-world scenarios demonstrate the superior effectiveness and efficiency of the COOPT framework across diverse datasets and architectures.

**Future Work.** In future work, we intend to (1) conduct an in-depth exploration of iterative data optimization through multiple rounds, enabling more dynamic collaboration among participants. (2) Furthermore, we aim to examine additional practical scenarios, such as participants optimizing their local data and then uploading it to the platform. This paradigm is anticipated to broaden the platform's functionalities, thereby increasing its applicability to real-world applications.

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
