# OpenReview forum: "Collaborative Data Optimization"
_ICLR.cc/2025/Conference — ICLR 2025 Conference Withdrawn Submission_

### Official Review · Reviewer_3gUg · 2024-10-20

**Soundness:** 2
**Presentation:** 3
**Contribution:** 2
**Rating:** 5
**Confidence:** 2

**Summary:**

The paper introduces COOPT, a collaborative, parallel framework designed to optimize data for deep learning in a highly efficient manner. The central idea is that participants can independently optimize subsets of data using pre-existing models (referred to as prior models), significantly reducing computational costs compared to sequential optimization. The framework addresses an important issue—Target Distribution Inconsistency—caused by using diverse prior models across participants. To counteract this, the paper proposes an alignment strategy to ensure consistency across the target distributions generated by different participants. Extensive experiments demonstrate COOPT’s effectiveness and efficiency, showing improvements in accuracy and training speed across various datasets and architectures.

**Strengths:**

- Originality: The proposed collaborative data optimization framework (COOPT) brings novelty by integrating the benefits of prior models for parallel data processing. This unique strategy of leveraging multiple independent optimizations underlines a creative combination of existing techniques such as knowledge distillation and dataset distillation. The paper presents a fresh approach to address computational challenges in large-scale, unlabeled datasets, which is significant in the current deep learning landscape.

- Quality: The experimental evaluation is thorough, covering multiple scenarios (e.g., with diverse datasets, architectures, and varying scales). The benchmarks and comparisons with state-of-the-art methods are well-chosen, supporting the claim that COOPT improves both training efficiency and model performance. The ablation studies and analysis of uniform value effectiveness add depth to the experimental section, illustrating the impact of the proposed alignment strategy.

- Clarity: The writing is clear and well-structured, with an explicit explanation of the challenges tackled (e.g., Target Distribution Inconsistency) and how COOPT overcomes them. Figures and tables are used effectively to summarize the results and support the claims.

- Significance: By improving training efficiency for large-scale data optimization, COOPT could have significant implications for both research and industrial applications, particularly in domains with vast amounts of unlabeled data. The potential to integrate it into open-source platforms could expand its real-world impact. The performance improvements achieved over existing methods, especially on larger datasets, underscore the practical value of the framework.

**Weaknesses:**

Complexity in Alignment Strategy: While the target alignment strategy effectively addresses the heterogeneity issue, the approach could become computationally expensive, particularly when aligning multiple participants with highly diverse prior models. The paper could benefit from discussing the potential trade-offs in more detail, including the computational cost of performing alignment versus the gains from collaborative optimization and versus the SSL methods conducted by a single participant.

Limited exploration of scalability: Although COOPT is designed to handle large-scale datasets, the experiments are primarily conducted on datasets like CIFAR and Tiny-ImageNet. It would strengthen the paper if larger datasets such as full ImageNet were used to demonstrate scalability more convincingly.

**Questions:**

It seems that the correlation of uniform value and performance are estimated based on SSL methods. Is it possible that the uniformity, as a desired property, is unique for SSL methods rather than other representation learning strategies? Could you provide me with more results on this correlation?

Are there any insights on how the framework would perform if all participants had low-quality or misaligned prior models? Would the alignment strategy still be effective in these cases?

The authors mention plans for continuous optimization in future work. Could they provide more details on how this would be implemented in practical scenarios, particularly if the participants update their models asynchronously?

Most importantly, I cannot understand why COOPT, or say aligning each representation space to the best one, could be better than learning representations of the whole dataset by the best prior representation model？It seems that there is no theoretical analysis explaining this relationship.

---

### Official Review · Reviewer_UYvw · 2024-11-01

**Soundness:** 1
**Presentation:** 1
**Contribution:** 1
**Rating:** 3
**Confidence:** 4

**Summary:**

The paper addresses the problems of data augmentation
and identifying smaller subsets of a dataset to accelerate
training without too much loss in accuracy of the eventually
learned model. The authors propose to split the dataset
into parts, apply different data augmentation and data
subsetting techniques to the parts, and then train
on their union. In experiments they compare their
method against using only one of the existing data augmentation
methods and show that they can learn better models
in less time.

**Strengths:**

- s1. combining different data augmentation techniques is interesting.
- s2. combining data augmentation techniques and subsetting
  techniques also might be interesting.

**Weaknesses:**

weak points.
- w1. the method is very simple.
- w2. the description of the method is overly complex.
- w3. the rationale for combining data augmentation and data subsetting
  techniques is not really clear.
- w4. the experiments are not fully clear.
- w5. the formalization of the problem is wrong.

review.

Researching the combination of different data augmentation
techniques is interesting. But the methods proposed by the paper
seems very simple, overly complex and the results unclear.
In more detail:

w1. the method is very simple.
w2. the description of the method is overly complex.
- can you describe your method in pseudocode?
- the most specific step, to align the pseudo targets of the
  different component methods, is only forward referenced
  in sec. 3.3, but it is not clear how it affects the overall method.

w3. the rationale for combining data augmentation and data subsetting
  techniques is not really clear.
- what evidence is there that combining both might be promising?
- how is data subsetting being used in your method? does it select
  subsets only from its own training data partition?

w4. the experiments are not fully clear.
- can you compare your approach to published results of an
  existing approach (referencing their tables) and clearly say which
  information the different approaches used?

w5. the formalization of the problem is wrong.
- a. def. 1 has many errors and loopholes:
  - do you only want to compare training losses? and why?
    usually one would be interested in validation losses here.
  - you likely mean that T' is given? for a general T' it cannot work,
    you could just choose T' := 0 or 1.
  - \Phi_{\theta} on the left and the right side of the equation denotes
    different models. The notation should make this clear.
- b. prop. 1 is not a proposition, it does not state a fact, but it defines
  what you later call "target distribution inconsistency".
  - this mismatch between the different output dimensionalities of
    the different component methods should be better introduced.
- c. def. 2 likely has a typo:
  - what is meant by "where (T^(i) and T^(j)) \in [0,1]" ? just a typo and "D_TV"
    is missing?
  - you later on never measure this quantity G.
  - to what extent is your re-alignment method guaranteeing to
    re-establish target distribution consistency? it seems just to reduce
    it somewhat.
- line 206: what does "O(|D|^2/K)^2" mean? (the last square sits outside of "O(...)".)

**Questions:**

- q1. How should combining data augmentation with data subsetting
  techniques help to learn better models or learn them faster?
  Is there any rationale that makes this plausible?
- q2. Can you compare your approach to published results of an
  existing approach (referencing their tables) and clearly say which
  information the different approaches used?

---

### Official Review · Reviewer_6TqA · 2024-11-03

**Soundness:** 2
**Presentation:** 2
**Contribution:** 2
**Rating:** 3
**Confidence:** 3

**Summary:**

The paper introduces CoOPT, a Collaborative Data Optimization framework for improving training efficiency in deep learning tasks. CoOPT enables multiple participants to independently optimize subsets of data in parallel, addressing inefficiencies in traditional data optimization methods that rely on sequential processes. The authors identify a key issue in their approach: inconsistencies in target distributions, and introduce an alignment strategy to improve consistency across the target distributions of all participants through the use of learnable transformation matrices. The experiments provided in the paper show CoOPT's superior efficiency over existing self-supervised and distillation-based methods.

**Strengths:**

1. Originality: CoOPT presents a new approach to collaborative data optimization by combining distributed optimization and a target alignment strategy minimizing inconsistencies across diverse participants. The proposed approach addresses significant bottlenecks associated with existing methods, improving both training efficiency and consistency across independently optimized data subsets.

2. Quality: The framework is validated with a series of experiments across multiple datasets and architectures, highlighting its flexibility and its suitability for collaborative settings characterized by heterogeneous data sources.

3. Clarity: The paper is generally well-organized, presenting its motivation, contributions, and methodologies in a structured manner. The experimental results show significant improvements in training efficiency, with notable speed gains over existing methods, making CoOPT suitable for scenarios where computational efficiency is prioritized.

4. Significance: The proposed approach shows potential for significantly reducing training costs and reliance on labeled data, which is particularly advantageous in distributed, collaborative environments where participants may vary widely in resources and data characteristics.

**Weaknesses:**

1. Lack of comparative analysis of alignment strategies: Although the paper introduces a target alignment strategy, it lacks a thorough comparison with alternative alignment or normalization techniques that could handle target inconsistencies. For example, domain adaptation approaches (such as source-free unsupervised domain adaptation by Tian et al., 2024) could potentially address similar issues, and comparing CoOPT's alignment strategy to these might reveal its unique advantages or limitations.

2. Conceptual clarity: While the paper is generally well-organized, with clearly defined sections for motivation, methodology, and experiments, the visual elements provided require further improvement. For instance, figure 1 could be enhanced in quality, figure 4d should show both the training and test accuracies, and figure 3c shows over 90% accuracy without any indication of why there is a sudden gap from figure 3b. Additionally, in Definition 2, the variable "I" is introduced without a prior definition, making the intended meaning unclear.

3. Scalability concerns: The computational and storage costs associated with the different stages of CoOPT are not thoroughly discussed. The authors should provide more insights into the computational complexity of the target alignment process relative to participant count and dataset size.

3. Theoretical justification: The approach's reliance on uniform value as a quality metric for prior models is supported mainly by limited empirical evidence (e.g., Figure 3c), yet the theoretical justifications of this metric in the context of data optimization remain vague. A more thorough theoretical discussion or derivation of why uniform value correlates with target quality would add depth to the method's rigour.

5. Experimental details: The paper would benefit from a clearer description of hyperparameters, hardware specifications (types and counts of GPUs/CPUs), experimental settings (e.g., participant counts in each experiment), and any additional configurations for implementing CoOPT. Moreover, some of the experimental results reported in the paper are unclear. For example, Table 4 shows comparisons across datasets rather than model architectures, and Figure 4d would be more informative if it included both training and test accuracies. The authors should revise the experimental results to improve clarity and correct errors, such as Table 2’s caption reading "four" instead of "three" datasets, an incomplete sentence in the experiments on shared data sizes ("However, increasing from 0.05% to 0.8%"), and Table 1’s subjective assessments, which would benefit from clearer criteria or more objective metrics.

6. Privacy concerns: The paper does not explicitly address or discuss privacy concerns, which could be a potential weakness in collaborative or distributed contexts where data privacy is crucial.

**Questions:**

1. As the number of participants increases, how does CoOPT manage potential increases in computational or storage costs, particularly for target alignment? Would larger participant counts introduce additional target inconsistencies, and if so, how might CoOPT address these?

2. Given that the current experiments use standard datasets, do the authors plan to apply CoOPT to more complex, high-dimensional, real-world data?

3. How does CoOPT handle scenarios where data distributions are highly imbalanced across participants? Would additional adjustments be necessary in the alignment strategy to maintain performance in such cases?

4. The experimental results focus on training efficiency, but how does CoOPT's efficiency compare in terms of memory usage or communication costs? Could these metrics also impact CoOPT's scalability in distributed environments?

**Details Of Ethics Concerns:**

CoOPT involves data subset distribution, target alignment, and model output sharing, which may expose sensitive information about the data or models used by participants.

---

### Official Review · Reviewer_EAzR · 2024-11-04

**Soundness:** 1
**Presentation:** 3
**Contribution:** 2
**Rating:** 3
**Confidence:** 4

**Summary:**

The paper focuses on an important and practical problem, i.e., data optimization. To solve this problem, a collaborative data optimization framework with better effectiveness and efficiency is proposed.

**Strengths:**

1. The paper is well-written and easy to understand.

2. The parallelized data optimization framework is interesting.

3. The experimental results outperform existing self-supervised learning methods in both effectiveness and efficiency.

**Weaknesses:**

1. The novelty is limited. The collaborative data optimization method with multiple participants makes sense and has potential practical value, but I don’t think it is a novel method.

2. The writing and organization is good. But the technique soundness is low, lacking of significant and in-depth technical contribution.

3. The theoretical analysis is missing. And many technical details are not well explained. For example, why choosing the uniform value loss for selecting prior model? It is better to add theoretical analysis about the target alignment since the alignment is very important in the proposed collaborative data optimization framework.

4. Many experimental details are not introduced. How many participants are used? How to split the unlabeled data for data optimization? How to process the scenarios where the input datasets are totally different from the prior datasets?

**Questions:**

see Weakness.

---

### Official Review · Reviewer_9YuC · 2024-11-04

**Soundness:** 2
**Presentation:** 3
**Contribution:** 3
**Rating:** 5
**Confidence:** 4

**Summary:**

This paper introduces an efficient and parallelized framework(CoOpt) for collaborative data optimization that allows participants to independently optimize data subsets. It first points out the computational efficiency shortcomings of previous data utilization Methods that operate sequentially, and then improves training efficiency by leveraging prior models to process data in parallel. The authors address the critical issue of Target Distribution Inconsistency arising from diverse prior models, by proposing an effective target alignment strategy. The authors demonstrate CoOpt's superior performance across various real-world scenarios and datasets with informative visualization support for effectiveness.

**Strengths:**

- Novel Approach: The proposed method introduces a parallel framework for collaborative data optimization, addressing the high time complexity associated with sequential optimization for the first time.
- Efficiency: Experimental results demonstrate that the CoOpt method significantly outperforms self-supervised learning optimization techniques across various datasets and model settings, while also exhibiting improved computational efficiency.
- Informative Experimental Results: The visualization presented in **Figure 4** illustrates the effectiveness of the alignment strategy and informs the design choices for alignment policies.

**Weaknesses:**

- Lack of Comparative Results: While **Table 1** summarizes several alternative data optimization methods, such as KD and DD, the experimental section lacks comparisons with these approaches. Additionally, the authors claim that the heavy costs associated with KD and DD stem from task-specific models. However, could using a pre-trained prior model on a larger dataset (like the ImageNet-1K mentioned in **Table 3**) effectively reduce data optimization costs?

**Questions:**

- Metric Used for Model Quality Evaluation: While there are some quantitative indicators demonstrating the correlation between model performance and the proposed uniform value, I am wondering the intuition behind selecting this metric over other model performance metrics(such as simple accuracy or mutual information).
- Effectiveness of Prior Models: I am also curious about why the task-agnostic prior model performs well with this method, even in the absence of task-specific knowledge. Is it due to a robust selection mechanism for the optimal prior model, or is it because the metric proposed in **Equation 4** has a strong correlation with model performance?
- Question on Results from **Table 4**: How do the results of the baseline method BYOL in **Table 4** reflect the diverse architectures of prior models?

Potential Typo: Is there a typo in the CF-100 BYOL baseline results in **Table 3** (51.7 ± 0.3)? I believe it should match the BYOL results in **Table 2** (51.7 ± 0.1) exactly.

---

### Note · Authors · 2024-11-26

I have read and agree with the venue's withdrawal policy on behalf of myself and my co-authors.